# Trans-Encapsidation of Foot-and-Mouth Disease Virus Genomes Facilitates Escape from Neutralizing Antibodies

**DOI:** 10.3390/v14061161

**Published:** 2022-05-27

**Authors:** Kay Childs, Ben Jackson, Yongjie Harvey, Julian Seago

**Affiliations:** The Pirbright Institute, Ash Road, Woking GU24 0NF, UK; kay.childs@pirbright.ac.uk (K.C.); ben.jackson@pirbright.ac.uk (B.J.); kitty.harvey@pirbright.ac.uk (Y.H.)

**Keywords:** foot-and-mouth disease, FMDV, co-infection, trans-encapsidation

## Abstract

Foot-and-mouth disease is an economically devastating disease of livestock caused by foot-and-mouth disease virus (FMDV). Vaccination is the most effective control measure in place to limit the spread of the disease; however, the success of vaccination campaigns is hampered by the antigenic diversity of FMDV and the rapid rate at which new strains emerge that escape pre-existing immunity. FMDV has seven distinct serotypes, and within each serotype are multiple strains that often induce little cross-protective immunity. The diversity of FMDV is a consequence of the high error rate of the RNA-dependent RNA polymerase, accompanied by extensive recombination between genomes during co-infection. Since multiple serotypes and strains co-circulate in regions where FMDV is endemic, co-infection is common, providing the conditions for recombination, and also for other events such as trans-encapsidation in which the genome of one virus is packaged into the capsid of the co-infecting virus. Here, we demonstrate that the co-infection of cells with two FMDVs of different serotypes results in trans-encapsidation of both viral genomes. Crucially, this facilitates the infection of new cells in the presence of neutralizing antibodies that recognize the capsid that is encoded by the packaged genome.

## 1. Introduction

Foot-and-mouth disease (FMD) is one of the most economically devastating diseases of livestock worldwide and presents a significant barrier to the international trade of animals and animal products. FMD is caused by foot-and-mouth disease virus (FMDV), an aphthovirus within the Picornaviridae family which is comprised of a single-stranded positive-sense RNA genome inside a non-enveloped icosahedral capsid. FMDV is highly contagious and can be spread through direct contact between animals; through contaminated feed, bedding, buildings, or vehicles; and via aerosols. The clinical signs of FMD include the appearance of blister-like sores (vesicles) on the mouth, hooves, and teats, in addition to fever, hypersalivation, reduced appetite and lameness. While adult animals typically recover, considerable morbidity is associated with infection and a loss of productivity as a result of reduced milk yields and weight loss. Infection also carries a high risk of death from myocarditis in neonatal animals (reviewed in [1,2]).

FMDV exists as seven distinct serotypes known as O, A, C, Asia-1, Southern African Territories (SAT) 1, SAT2, and SAT3. Serotype C has not been detected since 2004 and may be extinct in the field [3], but the remaining serotypes display overlapping regional distribution patterns across most of Africa, the Middle East, and parts of Asia where FMDV is currently endemic. Within each serotype there are multiple strains which can exhibit considerable genetic and antigenic diversity, which poses a challenge for the vaccination campaigns that are in place to control FMD. The ability of FMDV to undergo such rapid evolution is a result of error-prone replication of the genome and frequent recombination between different lineages, giving rise to novel combinations of viral sequences. In most FMD affected regions, multiple serotypes of FMDV co-circulate, and co-infection of a single animal with more than one serotype has been repeatedly documented [4,5,6,7,8]. Such co-infections are highly significant to the evolution of FMDV as they allow for the possibility of recombination between the co-infecting viruses and the generation of novel virus strains. The role of persistently infected carriers in the epidemiology of FMDV has been unclear, since it is generally considered that transmission from a carrier animal is unlikely (reviewed in [9]). However, the increased duration of infection, which can last many months, increases the probability of co-infection since carriers remain susceptible to subsequent infection with other serotypes. The potential of such superinfections to generate novel recombinants has recently been demonstrated in experimental co-infections of cattle [10]. In this study, cattle were infected with a second serotype of FMDV several weeks after the first, and evidence of inter-serotype recombination was found in multiple animals.

In terms of vaccine effectiveness, the production of viruses with novel recombinant capsids is of particular concern due to the possibility that they will partially or completely evade vaccine-induced immunity. The current commercially available vaccines against FMDV are inactivated virus vaccines formulated with an adjuvant, and protection is primarily mediated through the generation of neutralizing antibodies that recognize the capsid. One major antigenic site is located on the GH loop of the VP1 capsid protein which also contains the RGD (arginine, glycine, aspartic acid) motif that interacts with the integrin receptor to mediate cell entry [11,12,13,14,15]. Cryo-electron microscopy of neutralizing antibodies that are bound to virus particles has revealed that the mechanism of neutralization (at least for those antibodies studied) involves binding to the GH loop to block virus attachment to the receptor [16,17]. Hence, the composition of the capsid and its recognition by neutralizing antibodies is the key to immune escape by novel viruses. In addition to the production of genetic recombinants, co-infected cells can also produce chimeric viral particles containing capsid proteins from both parental viruses. The extent to which this can happen between serotypes is unclear due to the requirement for functional compatibility between subunits in the capsid, but it may be more likely to occur between different strains within the same serotype. Recently, we used recombinant serotype O FMDVs containing either an HA or a FLAG epitope tag in the VP1 capsid protein to demonstrate that, in co-infected cells, HA-VP1 and FLAG-VP1 proteins could assemble into the same capsid [18]. In theory, a capsid containing two different VP1 proteins may be less sensitive to neutralization by antisera against either one of the parental viruses. Furthermore, chimeric viruses containing the genome of one virus that is entirely encapsidated in the proteins that are encoded by the other co-infecting virus can be produced. This is known as trans-encapsidation but has also been referred to as “genomic masking” or “antigenic shielding”, as it may allow complete escape from neutralizing antibodies that recognize the capsid encoded by the packaged genome, enabling infection of an immune host. Trans-encapsidation has been studied extensively in poliovirus (PV) [19,20,21,22], but little is known about the process during FMDV co-infections.

Here, we have performed co-infections with different serotypes of FMDV and demonstrated that trans-encapsidation does occur, and that it enables escape from neutralizing antibodies in an in vitro system.

## 2. Materials and Methods

### 2.1. Cells and Viruses

ZZ-R 127 goat epithelium cells were maintained in Dulbecco’s modified Eagle’s medium/Ham’s F12 medium (Merck, Gillingham, UK) with 10% fetal bovine serum (FBS) and penicillin/streptomycin (Merck, Gillingham, UK) at 37 °C in 5% CO_2_. The virus strains used in this study are FMDV O/ETH/29/2008 (FMDV-O); FMDV A/ETH/9/2008 (FMDV-A); FMDV SAT1/KEN/80/2010 (FMDV-SAT1); and FMDV SAT2/ETH/65/2009 (FMDV-SAT2) [23]. These viruses are all wild type and non-cell culture adapted but caused a complete cytopathic effect (CPE) of ZZ-R 127 due to their expression of the key FMDV receptor integrin αvβ6 [24].

### 2.2. Antisera and In Vitro Neutralisation

Monovalent vaccinate sera were collected on day 21 from cattle that had been vaccinated on day 0 and boosted on day 14 with inactivated, adjuvanted preparations of either FMDV O/ETH/29/2008; FMDV A/ETH/9/2008; FMDV SAT1/KEN/80/2010; or FMDV SAT2/ETH/65/2009 as described in [23]. All four serum samples had neutralizing antibody titers > 2 log_10_ (considered protective) as determined by homologous virus neutralization test (VNT) [23]. For each in vitro neutralization, 75 µL of respective serum (with a reciprocal neutralizing titer of 355 determined using BHK-21 cells) and 5 µL of virus (at a concentration of 6.5 log_10_ PFU/mL) were pre-incubated for 1 h at 37 °C prior to addition to ZZ-R 127 cells. Three hours later, the cells were washed three times with medium and fresh medium was then added. Sixteen hpi cell cultures were freeze-thawed once, clarified at 1600× *g* for 15 min and the collected supernatant was subjected to RNA isolation and RT-PCR analysis.

### 2.3. Plaque Assays

Ten-fold virus dilutions were used to infect triplicate wells of confluent ZZ-R 127 cells that were pre-seeded in 6-well tissue culture plates. Following adsorption at 37 °C for 1 h, the inoculum was removed and 2 mL of indubiose (MP Biomedicals, Santa Ana, CA, USA) overlay was added. After a further 48 h incubation at 37 °C, the cells were fixed by the addition of 10% tetrachloroauric acid (Merck, Gillingham, UK) for 30 min. Indubiose plugs were then removed, and the cells were stained with methyl blue solution (PBS, 10% formaldehyde, 10% of 1% methyl blue in ethanol) prior to the determination of the plaque forming units/mL (PFU/mL).

### 2.4. RNA Isolation and RT-PCR

RNA was prepared using a QIAamp Viral RNA Mini Kit (Qiagen, Manchester, UK), and reverse transcribed and amplified using QIAGEN OneStep RT-PCR Kit (Qiagen, Manchester, UK) with the following serotype-specific primers: O-Forward (5′-CCTACCACAAGGCACCAC-3′) plus O-Reverse (5′-CCAACACTTGGAGATCG-3′), 138 bp; A-Forward (5′-CTCTCCAGAACGCGAGC-3′) plus A-Reverse (5′-TGGATGGTCGTGGCTCT-3′), 229 bp; SAT1-Forward (5′-GGTACGAACAAGTGGGTTGG-3′) plus SAT1-Reverse (5′-TCCGCCTCCGTGTAGATCC-3′), 290 bp; SAT2-Forward (5′-GCAGCACACACGTGTAC-3′) plus SAT2-Reverse (5′-GAAACCAAAGTTGAAGGTGC-3′), 395 bp.

## 3. Results

### 3.1. Establishment of a Serotype-Specific PCR Assay

To enable the specific detection of individual serotypes from the mixed infections of FMDV-O, FMDV-A, FMDV-SAT1, and FMDV-SAT2, unique PCR primer pairs were designed to amplify a fragment encoding part of the VP1 capsid protein. To confirm their specificity, infectious clone plasmids for each virus were combined as shown in Figure 1a and subjected to PCR analysis with each of the four primer sets. Mix 1 contained all four plasmids and mixes 2–5 each had one virus omitted. As expected, PCR products were obtained from mix 1 using all four primer sets, whereas no PCR product was obtained from the mixes in which the target serotype plasmid was not present (Figure 1b–e).

### 3.2. Trans-Encapsidation between O and A Serotype Viruses Enables Escape from Neutralising Antibodies

We hypothesized that cells co-infected with two viruses of different serotypes would produce trans-encapsidated progeny viruses, and that this would allow the genome of one serotype to be carried in the capsid of the other serotype into new cells in the presence of neutralizing antibodies. To test this, goat epithelium cells were simultaneously co-infected with an O serotype and an A serotype FMDV at an MOI of 1. Simultaneous co-infections were carried out as we have previously observed variation in virus yields following staggered infections in vitro, presumably due to superinfection exclusion. The supernatant containing the progeny viruses was collected 16 h post-infection and incubated with anti-O antiserum, anti-A antiserum or both for 1 h prior to infection (MOI 0.05) of a fresh monolayer of goat epithelium cells. At 3 h following infection, these cells were washed to remove virus/antibody complexes that had not gained entry, and 16 h post-infection the presence of both O and A serotype genomes in the second round of infections was determined by serotype-specific RT-PCR (shown schematically in Figure 2a). In parallel control experiments, FMDV-O and FMDV-A viruses were grown separately and subsequently combined before incubation with the antisera, infection (MOI 0.05) of fresh cells and RT-PCR analysis.

Following the first round of infection, viruses that were obtained from the co-infection and viruses that were grown in separate cultures exhibited comparable titers of 6–7 log_10_ PFU/mL (Figure 2b), and yielded RT-PCR products with the appropriate O and A-specific primers (Figure 2c). In the samples in which FMDV-O and FMDV-A were grown separately and then combined, incubation with anti-O antiserum prevented infection of the cells with FMDV-O and no obvious PCR product was obtained with the O-specific PCR primers, whereas infection with FMDV-A was unaffected and an A-specific RT-PCR product was observed (Figure 2d). Conversely, incubation with anti-A antiserum blocked infection with FMDV-A, but not FMDV-O. Incubation with anti-O and anti-A antisera together blocked infection with both viruses and no obvious RT-PCR products were detected with either O or A-specific primer sets (Figure 2d). In contrast, using the samples taken from the co-infection, incubation with anti-O antiserum failed to prevent introduction of the FMDV-O genome into cells in the next round of infection (Figure 2e). However, incubation with both anti-O and anti-A antisera almost completely prevented the detection of the FMDV-O genome in the subsequent infection, indicating that the FMDV-O genome entered cells in a capsid that can be neutralized by anti-A antiserum. Similarly, incubation with anti-A antiserum did not block the delivery of the FMDV-A genome into the next batch of cells, whereas the combination of both anti-A and anti-O antisera was successful. In the absence of antisera, both FMDV-O and FMDV-A from either the co-infection or the separate infections combined successfully infected cells (Figure 2f).

These data show that the genomes of FMDV-O and FMDV-A can be trans-encapsidated by the capsids of the other virus during co-infection, and that this overcomes the ability of serotype-specific antibodies to block the delivery of that genome into naïve cells.

To confirm virus neutralization, we carried out simultaneous microscopic observation of cells to monitor the development of cytopathic effect (CPE). Infection of the cells with viruses obtained from the co-infection experiment produced complete CPE within 16 h in the absence of neutralizing antisera (Appendix A). Pre-incubation with anti-O or anti-A antisera (individually) partially blocked the development of CPE, whereas no CPE was observed at 16 hpi in the cells that were infected with viruses pre-incubated with both anti-O and anti-A antisera. Of note, barely detectable background levels of RT-PCR products were observed and were possibly due to incomplete neutralization leading to a very low level of infection (Figure 2d,e).

### 3.3. Trans-Encapsidation between SAT1 and SAT2 Serotype Viruses Enables Escape from Neutralising Antibodies

To see if the trans-encapsidation that was observed for serotypes A and O was a general phenomenon, we sought to confirm this result using other serotypes of FMDV. Hence, a similar experiment was conducted in which the cells were individually infected or co-infected with FMDV-SAT1 and FMDV-SAT2. The viruses that were produced from these cultures reached similar titers of around 7 log_10_ PFU/mL (Figure 3a) and yielded RT-PCR products with the appropriate SAT1 and SAT2-specific primers (Figure 3b). Analysis of the samples in which FMDV-SAT1 and FMDV-SAT2 were grown in separate cultures and then combined showed that incubation with anti-SAT1 antiserum almost completely blocked the detection of the SAT1 genome in the subsequent round of infection but did not prevent the detection of SAT2 (Figure 3c). Similarly, incubation with anti-SAT2 antiserum blocked subsequent infection with FMDV-SAT2, but not FMDV-SAT1. As expected, neither FMDV-SAT1 nor FMDV-SAT2 were detected in the cells that were exposed to viruses that were incubated with both anti-SAT1 and anti-SAT2 antisera together (Figure 3c).

Consistent with the above described trans-encapsidation results, incubation of viruses that were obtained from the FMDV-SAT1 and FMDV-SAT2 co-infection with anti-SAT1 antiserum did not prevent transfer of the FMDV-SAT1 genome to the cells in the next round of infection, whereas incubation with both anti-SAT1 and anti-SAT2 antisera resulted in no detectable SAT1 RNA (Figure 3d). In the reciprocal experiment, incubation with anti-SAT2 antiserum did not prevent transfer of the FMDV-SAT2 genome into the cells, but incubation with both anti-SAT1 and anti-SAT2 antisera blocked it completely (Figure 3d). In the absence of antisera, both FMDV-SAT1 and FMDV-SAT2 from either the co-infection or the separate infections combined all successfully infected cells (Figure 3e). Observation of the cell cultures showed that viruses that were produced from the FMDV-SAT1 and FMDV-SAT2 co-infection caused complete CPE at 16 hpi, and that incubation with either anti-SAT1 or anti-SAT2 antisera provided partial protection, whereas incubation with both antisera provided complete protection from CPE (Appendix A).

## 4. Discussion

One of the most significant challenges facing efforts to control FMD is the vast antigenic heterogeneity within viral populations as a result of the high mutation rate and rapid evolution of FMDV. New variants that are poorly cross-neutralized by antibodies induced by prior infection or vaccination continually emerge and replace existing lineages, causing waves of infection to move through susceptible populations. It is therefore necessary to maintain a continual program of surveillance and to update vaccine strains regularly to prevent outbreaks, and a more thorough understanding of the mechanisms involved in driving antigenic diversity and immune escape is critical to developing new strategies to combat the spread of FMD.

Co-infection occurs regularly in the field and contributes significantly to the evolution of FMDV, as it provides the conditions under which recombination can occur. Furthermore, it enables the mixing of components from the co-infecting viruses and the generation of chimeric capsids and trans-encapsidation of genomes. Much of our understanding of trans-encapsidation in picornaviruses comes from studies on PV. Early co-infection experiments with different subtypes of PV showed that trans-encapsidation could generate viruses with an antibody susceptibility that differed from their genotype [25,26]. Later work with PV replicons revealed that although trans-encapsidation by the capsid proteins from the homologous virus was by far the most efficient, heterologous trans-encapsidation was observed with variable efficiencies depending on the origin of the capsid proteins used [19,20,21]. Interestingly, it was not only the capsid proteins from other PV subtypes that could successfully package the replicon, but also the capsid proteins from other genera of picornaviruses such as enterovirus and coxsackievirus, albeit with reduced efficiency [20]. Examples of such cross-genera trans-encapsidation events have also been found in vivo, including a virus that was comprised of an FMDV genome inside a bovine enterovirus (BEV) capsid [27].

Replicon RNA which is transfected or electroporated into cells is a less efficient substrate for encapsidation than RNA that is delivered in an infectious particle. The reason for this is unclear but may be due to differences in subcellular localization and the fact that the packaging of the genome is mechanistically linked to RNA replication and successful translation [22]. However, when homologous capsid proteins are provided in trans they are still more efficient at encapsidating replicon RNA than heterologous capsid proteins, indicating that there is some degree of specificity in the selection of the RNA to be packaged. In a co-infection, the two viruses might be replicating in different replication complexes and trans-encapsidation requires the mixing or transfer of components between complexes. Therefore, it may be preferable to study trans-encapsidation events during bona-fide co-infections rather than in replicon systems.

We recently showed that the co-infection of cells with two different FMDV viruses can be studied in vitro using viruses with epitope tags in their VP1 capsid proteins [18]. These experiments showed for the first time that co-infection can result in the trans-encapsidation of FMDV genomes, and that chimeric FMDV capsids containing protein subunits from both parental viruses can be generated. These events have important implications for immune evasion in vivo. Here, we take these observations further and show that trans-encapsidation can occur between wild-type FMDVs of different serotypes. Importantly, we demonstrate that this facilitates escape from neutralizing antibodies and allows the transmission of a genome into new cells in the presence of neutralizing antiserum. These data have clear relevance for the evolution of FMDV and vaccine effectiveness in the field.

## Figures and Tables

**Figure 1 viruses-14-01161-f001:**
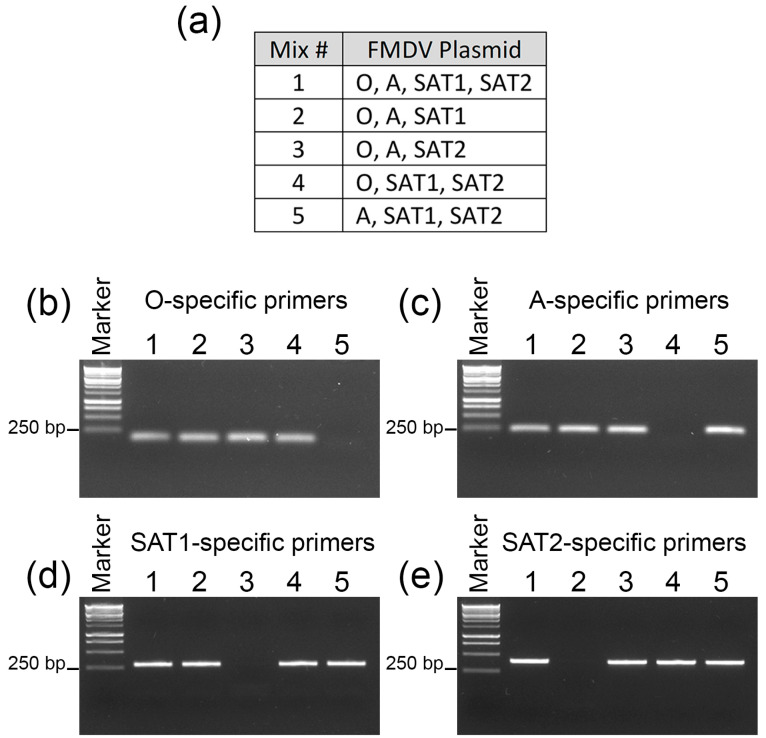
Establishment of a serotype-specific PCR assay. (**a**) Plasmids containing infectious DNA copies of the FMDV-O, FMDV-A, FMDV-SAT1, and FMDV-SAT2 genomes were mixed in equimolar amounts in the indicated combinations. Mixtures of plasmids were subjected to PCR analysis using primer sets specific for FMDV-O (**b**), FMDV-A (**c**), FMDV-SAT1 (**d**), or FMDV-SAT2 (**e**).

**Figure 2 viruses-14-01161-f002:**
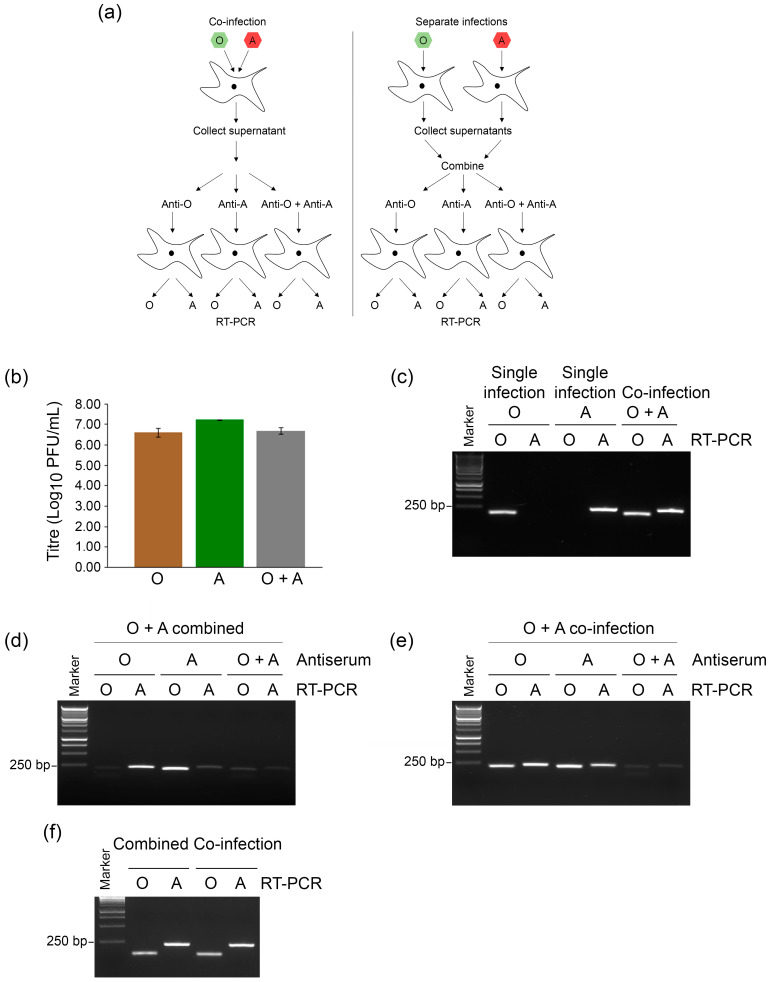
Trans-encapsidation between O and A serotype viruses enables escape from neutralizing antibodies. (**a**) Schematic diagram of the experimental workflow. ZZ-R 127 cells were either co-infected with FMDV-O and FMDV-A at an MOI of 1 per virus or infected with each virus separately for 16 h. Supernatants were collected, and those from the two separate infections were combined in a single tube. Aliquots of these samples were then incubated in the presence of anti-O, anti-A, or anti-O + anti-A antisera for 1 h prior to infection (MOI 0.05) of a fresh monolayer of ZZ-R 127 cells. After 16 h, RNA was prepared from the cells and RT-PCR was performed using primers specific for FMDV-O or FMDV-A. (**b**) Viruses obtained from the separate infections and the co-infection were titered by plaque assay. (**c**) Viruses obtained from the separate infections and the co-infection were subjected to RT-PCR analysis using FMDV-O or FMDV-A specific primer sets to confirm the presence of both genomes prior to incubation with antisera. (**d**) RT-PCR analysis of RNA purified from cells exposed to viruses obtained from separate combined infections incubated with either anti-O, anti-A, or anti-O plus anti-A antisera using O or A specific primers. (**e**) RT-PCR analysis of RNA purified from cells exposed to viruses from the co-infection incubated with either anti-O, anti-A, or anti-O plus anti-A antisera using O or A specific primers. (**f**) RT-PCR analysis of RNA from cells exposed to viruses incubated without antisera.

**Figure 3 viruses-14-01161-f003:**
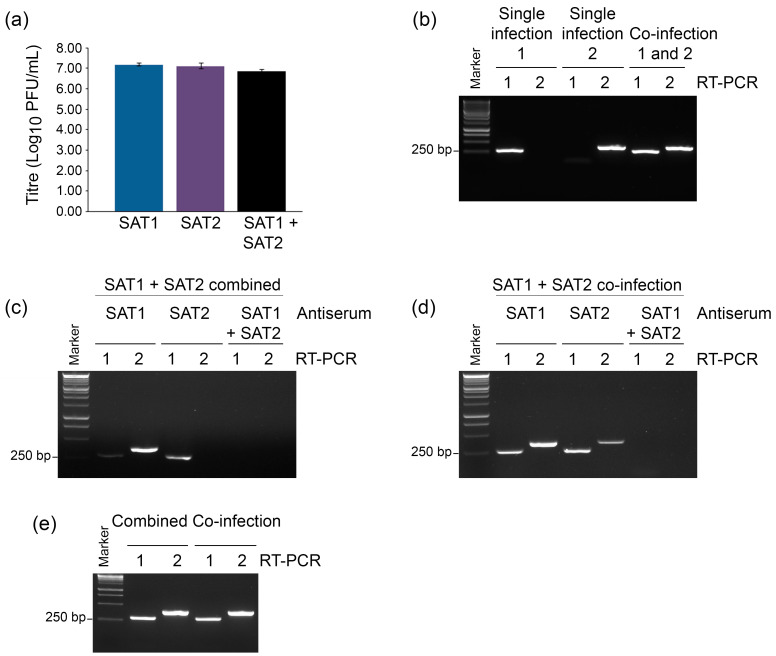
Trans-encapsidation between SAT1 and SAT2 serotype viruses enables escape from neutralizing antibodies. ZZ-R 127 cells were either co-infected with FMDV-SAT1 and FMDV-SAT2 or infected with each virus separately and processed as described in Figure 2a. (**a**) Viruses obtained from the separate infections and the co-infection were titered by plaque assay. (**b**) Viruses obtained from the separate infections and combined were subjected to RT-PCR analysis along with the products of the co-infection using FMDV-SAT1 (1) or FMDV-SAT2 (2) specific primer sets to confirm the presence of both genomes prior to incubation with antisera. (**c**) RT-PCR analysis of RNA purified from cells exposed to viruses obtained from the separate combined infections and incubated with anti-SAT1, anti-SAT2, or anti-SAT1 + anti-SAT2 antisera using SAT1 or SAT2 specific primers. (**d**) RT-PCR analysis of RNA purified from cells exposed to viruses from the co-infection incubated with anti-SAT1, anti-SAT2, or anti-SAT1 + anti-SAT2 antisera using SAT1 or SAT2 specific primers. (**e**) RT-PCR analysis of RNA from cells exposed to viruses incubated without antisera.

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
