# Peer review of "Trans-Encapsidation of Foot-and-Mouth Disease Virus Genomes Facilitates Escape from Neutralizing Antibodies"

_viruses, 2022, doi:10.3390/v14061161_

Round 1

Reviewer 1 Report

In their manuscript entitled “Trans-encapsidation of foot-and-mouth disease virus genomes facilitates escape from neutralising antibodies.” (manuscript ID: viruses-1680640) Childs et. al. present experimental data extending a previous study by the group in which chimeric viruses were used to observe trans-encapsidation of FMDV genomes. The group has a strong track-record on FMDV research and the methodology used here is the appropriate for this type of studies.

Although it is somehow expected that the same phenomenon observed before with tagged viruses would be occurring with wild-type viruses therefore lacking novelty, data presented here is interesting and contributes to the field.

Specific comments:

  1. Line 18; it is stated here that trans-encapsidation shields the genome from neutralising antibodies. This sentence must be re-worded as it can be wrongly interpreted that neutralising antibody could directly bind to viral RNA.
  2. Line 80-81; same as above this sentence needs to be made clearer to avoid any misinterpretation.
  3. Figure 2 and Figure 3; the authors did synchronous infections with 2 different serotypes. What happens in the case of asynchronous infection? Is superinfection exclusion observed in the early phases of in vitro infection and therefore if the two infections were happening sequentially the outcome would be different? Authors must add a few clarifying sentences in the corresponding text.
  4. Figure 2d, 2e and S1; it appears that virus neutralisation was incomplete as faint bands of the same size as the brighter bands for FMDV-O and FMDV-A can be seen on these panels in the presence of antibody (a double band for serotype O in the case of co-infection and when the supernatant was treated with both anti-O and anti-A antisera). The authors suggest that the data in Figure S1 demonstrates complete neutralisation of viruses pre-incubated with both anti-O and anti-A antisera. How is the presence of nucleic acid explained? Could it be that the lack of CPE at 16 hpi is due to the fact that the remaining non-neutralised virus was at a very low level and if the infection was allowed to progress for longer CPE would be evident?
  5. In contrast the picture is much clearer in Figure 3 and no bands can be seen in panel 3d (SAT1+SAT2 co-infection and incubation with anti-SAT1 and anti-SAT2 antisera). Interestingly a faint band of the correct size can be seen in panel 3c suggesting that neutralisation of SAT1 was not complete on this experiment.
  6. Although standard PCR followed by gel electrophoresis is a valid method for this kind of studies it would have been better to utilise qPCR and quantify viral RNA copy number. This would make the results clearer and the respective section easier to follow. Furthermore, in the provided file with the original gel photos it appears that the exposure time used to capture the different images was not always the same. It would be good to amend this as it will make comparison easier.

Minor points:

  1. What is the size of the PCR bands? This should be included in the materials and methods section.
  2. Figure 2 and Figure 3 are a bit untidy and should be done to a higher standard.
  3. As the supplementary figures are referenced in the text add scale bars must be added in the different panels. Also conditions of infections (MOI and timepoint) must be included on the figure legends.

Reviewer 2 Report

This manuscript is relevant to the FMD field and describes well a mechanism of trans-encapsidation, relevant to virus evolution and disease epidemiology.

There is a lack of detail in some of the methodology, pointed out in the attached pdf comments.

If these details are included the manuscript should be good for publication.
